# The Association of Attending Physicians’ Publications and Patients’ Readmission Rates: Evidence from Tertiary Hospitals in China Using a Retrospective Data Analysis

**DOI:** 10.3390/ijerph19159760

**Published:** 2022-08-08

**Authors:** Menghan Shen, Xiaoxia Liang, Linyan Li, Yushan Wu, Yuanfan Yang, Raphael Zingg

**Affiliations:** 1Center for Chinese Public Administration Research, School of Government, Sun Yat-sen University, Guangzhou 510275, China; 2School of Data Science, City University of Hong Kong, Kowloon Tong, Hong Kong 999077, China; 3Department of Infectious Diseases and Public Health, Jockey Club College of Veterinary Medicine and Life Sciences, City University of Hong Kong, Kowloon Tong, Hong Kong 999077, China; 4Department of Environmental Health, Harvard T.H. Chan School of Public Health, Boston, MA 02215, USA; 5The Jockey Club School of Public Health and Primary Care, Chinese University of Hong Kong, N.T. HKSAR, Shatin, Hong Kong 999077, China; 6Department of Neurosurgery, University of Alabama at Birmingham, Birmingham, AL 35294, USA; 7Waseda Institute for Advanced Study, Waseda University, Tokyo 169-0051, Japan; 8ETH Zurich, Center for Law & Economics, 8029 Zurich, Switzerland

**Keywords:** China, publication, physician, research, readmission rates

## Abstract

Background: Physicians play a unique role in scientific and clinical research, which is the cornerstone of evidence-based medical practice. In China, tertiary public hospitals link promotions and bonuses with publications. However, the weight placed on research in the clinician’s evaluation process and its potential impact on clinical practice have come under controversy. Despite the heated debate about physicians’ role in research, there is little empirical evidence about the relationship between physicians’ publications and their clinical outcomes. Method: This paper examines the association of the quantity and quality of tertiary hospitals’ attending physicians’ publications and inpatient readmission rates in China. We analyzed a 20% random sample of inpatient data from the Urban Employee Basic Medical Health Insurance scheme in one of the largest cities in China from January 2018 through October 2019. We assessed the relationship between the quantity and impact factor of physicians’ publications and 30-day inpatient readmission rates using logistic regression. There were 111,965 hospitalizations treated by 5794 physicians in our sample. Results: Having any first-author publications was not associated with the rate of readmission. Among internists, having clinical studies published in journals with an average impact factor of 3 or above was associated with lower readmission rates (OR = 0.849; 95% CI (0.740, 0.975)), but having basic science studies published in journals with an average impact factor of 3 or above was not associated with the rate of readmission. Among surgeons, having clinical studies published in journals with an average impact factor of 3 or above was likewise associated with lower readmission rates (OR = 0.708 (0.531, 0.946)), but having basic science studies published in journals with an average impact factor of 3 or above was associated with higher readmission rates (OR = 1.230 (1.051, 1.439)).

## 1. Introduction

Clinical research is the cornerstone of evidence-based medical practice [1,2,3]. Published research studies, especially landmark trials, have resulted in major improvements in the prevention and treatment of diseases [4,5,6,7]. Excellence in research for physicians is of particular importance. Indeed, many major developments in modern medicine have been driven by physicians looking for a cure of disease [8]. Physicians bring a unique perspective to research, as their scientific questions arise while taking care of patients [4]. Physicians can actively contribute to the pursuit of knowledge by bringing clinical needs into research [5,6]. They can also apply research findings to clinical practice and may be more effective in communicating clinical and translational research findings to patients and the general public than those who do not conduct research [5,6,9,10].

China’s large population and heavy disease burden create challenges for researchers but also offer unique opportunities to study disease management [11,12,13,14]. To promote research, Chinese teaching hospitals affiliated with medical schools now require scientific publications in Science Citation Index journals when recruiting health care providers and have linked promotions and bonuses to new publications [15,16,17,18]. The financial bonuses are typically linked to the journal’s impact factor to provide incentive for higher-impact research and publication [15,18,19]. The model, together with the growth of research funding in China, has placed a strong emphasis on medical research during practice. Recently, however, the weight placed on research in the clinician evaluation process and its potential impact on clinical practice have come under scrutiny [16].

Proponents of physicians’ involvement in scientific research suggest that it enables them to evaluate their own practices in a robust, scientific manner [8], which should lead to better clinical outcomes [20]. Survey findings indicate that physicians believe it is necessary for them to conduct scientific research [21]. Detractors argue that the majority of clinicians should focus on their clinical practice rather than spending time seeking grants and publications. Heavy clinical workloads, a lack of time and energy, and a lack of formal training in scientific writing are other obstacles to physicians’ participation in research [22]. Thus, the present medical system is in urgent need of reform [23]. This is especially true in countries such as China, with a large population and considerable medical needs. A 2014 national survey showed that over 90% of doctors in tertiary hospitals need to work overtime to handle their caseload [24,25]. Requiring or incentivizing physicians to conduct research may contribute to physician burnout by making it harder for physicians to balance work and family responsibilities. Finally, some detractors argue that promotions should be given to physicians with the most experience in patient treatment, not those with the most scientific success.

Despite the heated debate about physicians’ role in research, there is little empirical evidence about the relationship between physicians’ publications and their clinical outcomes. To address this gap in the literature, using data from one of the largest Chinese cities, this paper examines the association between physicians’ publication records and their association with 30-day inpatient readmission rates, a common measure of clinical outcomes [26,27,28,29].

## 2. Materials and Methods

### 2.1. Data

We analyzed a 20% random sample of inpatient data from the Urban Employee Basic Medical Health Insurance (UEBMI) scheme in one of the cities in China from January 2018 through October 2019. The sample was drawn randomly at the individual level. The city is one of the largest and most developed cities in China, with top-tier healthcare resources. Enrollment in the UEBMI is mandatory for anyone employed in the formal sector and voluntary for those who are self-employed. It is part of the social health insurance program that covers over 95% of the population in China [30]. Our dataset contains information on enrollees’ birth year, gender, residence, and work position as well as their insurance payment information. In addition, the dataset contains healthcare utilization information, including the time of each visit, a breakdown of costs, the name of the attending physician and the department, the hospital name, and the patient’s primary diagnosis based on International Classification of Diseases (ICD-10) codes at the visit level.

In the dataset, 90% of the visits have the name of the attending physician. We obtained data on the physicians’ publications by collecting their hospitals’ publication history from *PubMed*. Then, we merged information on physicians and publications using the first and last names of the physician and the name of their institution. There were over 99% unique Chinese name and hospital pairs in our setting. In cases where physicians with the same name were working in the same hospital, we manually assigned the physicians by their department to ensure matching accuracy.

To determine journals’ impact factor, we matched journal names from the *PubMed* database with impact factors from the *Web of Science*. Physicians who could not be matched with a name and affiliation were considered not to have published in any international journals indexed by *PubMed*. We manually checked the match accuracy for 200 randomly selected physicians and found that our results were consistent with those from the method described above. We assessed only English-language publications because the promotion of physicians in China is largely dependent on their publications in English-language journals. Please see the flow chart for constructing the data in Appendix A.

### 2.2. Inclusion and Exclusion Criteria

We restricted our analysis to visits by individuals who were continuously enrolled in the UEBMI during the period of study. We included inpatient visits at all levels when we calculated readmission rates. We excluded visits that took place in intensive care units. We also excluded patients who left against medical advice and those with cancer or mental illness, as their inpatient visits are recurring. We excluded visits in which patients were discharged in September or October of 2019 in order to ensure we had sufficient follow-up information on patients’ readmission.

We restricted our analysis to attending physicians at tertiary hospitals that are also teaching hospitals, as they are expected to perform high-quality research and clinical practice at the same time. There were 38 teaching hospitals in our sample, which account for all of the tertiary teaching hospitals in the city. We restricted our analysis to attending physicians who were designated as attending patients in the claims data. The attending physicians in these hospitals generally have similar levels of work experience and education. In total, 4011 internists and 1783 surgeons were included in our dataset.

### 2.3. Exposure Variables

We focused on attending physicians because they are responsible for making treatment decisions. In addition, as attending physicians in teaching universities, they generally have similar years in training and experiences. We examined physicians in internal and surgical departments separately. We analyzed only first-author publications in international journals from January 2010 to December 2019. We analyzed only first-author publications because, in China, the evaluation system usually only recognizes publications where someone is the first author or the corresponding author. The first author is usually responsible for the primary bulk of the work, while the corresponding author is generally considered as the senior in the teamwork. We divided the publications according to whether they reported on clinical studies (i.e., where the main subject matter was clinical medicine) or basic science studies (where the main subject matter was not clinical medicine) [4] as well as by impact factor. We looked at impact factors of 3+ and 5+ because these impact levels are usually considered in job-promotion criteria. The exposure variables were whether physicians had any first-author publications, any clinical studies, and any basic science studies in international journals indexed by *PubMed*; within each of these categories, we also looked specifically at whether physicians have research papers published in journals with an average impact factor of 3+ or 5.

### 2.4. Outcome Variables

We analyzed 30-day and 60-day readmission rates using claims data. A large number of studies have used this measure as an index of clinical outcomes [26,27,29,31]. Hospital readmissions are associated with high direct and indirect costs to families, the healthcare system, and society [32] but are sometimes avoidable [33]. Increasingly, hospital readmission rates are used for both quality improvement and cost control [34]. Thus, we adopted readmission rates as our outcome variable.

### 2.5. Adjustment Variables

We adjusted for patient characteristics, including age, age squared, gender, primary diagnosis, number of chronic diseases, number of outpatient and inpatient visits from January 2015 to December 2017, type of residence (urban vs. rural), bureau of residence, and employment status (working in the formal vs. informal sector). We also adjusted for the primary diagnosis coded for each visit, the department of the visit, and whether the visit was an emergency visit, and we included indicators for the month, year, and day of the week the visit took place.

### 2.6. Statistical Analysis

We examined the association between physicians’ publications and 30-day inpatient readmission rates (whether patients were readmitted within 30 days of discharge) in any healthcare setting using logistic regression as our outcome is a binary variable. We clustered standard errors at the physician level to account for the correlation of visits to the same physician. In Model 1, we controlled for all of the patient information discussed above. In Model 2, we further controlled for hospital fixed effects. We considered the results from both models because it is possible that there are knowledge spillover effects within the same hospital. In Model 2, controlling for hospital fixed effects may reduce concerns about omitted variable bias correlated with physicians’ publications and the quality of the hospital but may be downward-biased because of positive externality within the hospital.

We analyzed potential mechanisms producing differences between physicians by including differences in the length of stay, patient volume (number of hospitalized patients each physician treated monthly), and patient costs. As a sensitivity analysis, we used a multivariable probability model as well as examining 60-day readmission rates (readmissions that took place within 60 days). Data analyses were conducted using Stata, version 16 (StataCorp).

## 3. Results

Appendix A presents descriptive statistics on physicians’ first-author publications in the past five years. The information presented in the table is the percentage of physicians who have published as the first author and the different levels of journals. The table suggests that 37.77% of internists and 41.22% of surgeons had at least one first-author publication in the past five years. Among internists, 17.65% published in journals with an average of impact factor of 3 or above, and 4.34% published in journals with an average impact factor of 5 or above. For surgeons, the corresponding percentages were 16.10% and 3.70%, respectively.

Appendix A presents descriptive statistics on patient characteristics at the visit level. The table presents mean and standard deviation information on age, length of stay, and number of outpatient visits. We present percentage information for gender, residence, position, and number of chronic diseases.

The mean age of internists’ patients was 59.33 (SD = 16.91), and that of surgeons’ patients was 56.47 (SD = 17.37). On average, patients were hospitalized for 9.06 days (SD = 6.81) and 7.75 days (SD = 6.91) for internists and surgeons, respectively; the number of outpatient visits for the two types of patients was 23.67 (SD = 25.85) and 21.86 (SD = 25.59). Female patients accounted for 52.53% of internists’ patients and 50.09% of surgeons’ patients. Among internists’ patients, 13.05% had a rural residence, and 60.39% of them worked as a worker. Among surgeons’ patients, 16.26% had a rural residence, and 61.77% of them worked as a worker. Among internists’ patients, 21.73% had one chronic disease, and 38.37% had multiple chronic diseases. Among surgeons’ patients, these percentages were 19.72% and 25.11%, respectively. In total, 84,954 patients were treated by internists, and 27,011 patients were treated by surgeons.

Table 1 and Table 2 present the results of regressions examining the association between doctors’ first-author publications and patients’ 30-day readmission rates. In the tables that present the results, we first present the percentage of readmission rates by doctors’ publication status, followed by adjusted odds ratios calculated using Model 1 and Model 2. In both Models 1 and 2, having any publication, any clinical study, or any basic science study in an international journal is not associated with differences in readmission rates for either internists or surgeons.

As shown in Table 1, for internists, in Model 1, having a publication in journals with an average impact factor of 3 or above was statistically significantly associated with lower 30-day patient readmission rates (OR = 0.917; 95% CI (0.846, 0.994)), but this relationship was not statistically significant in Model 2. In both models, having a publication in journals with an average impact factor of 5 or above (Model 1: OR = 0.819 (0.716, 0.936); Model 2: OR = 0.859 (0.760, 0.972)), having published clinical studies in journals with an average impact factor of 3 or above (Model 1: OR = 0.786 (0.683, 0.905); Model 2: OR = 0.849 (0.740, 0.975)), and having published clinical studies in journals with an average impact factor of 5 or above (Model 1: OR = 0.699 (0.570, 0.858); Model 2: OR = 0.746 (0.602, 0.926)) were statistically significantly associated with lower 30-day readmission rates. Having basic science publications was not statistically significantly associated with lower 30-day readmission rates regardless of journals’ impact factor.

As shown in Table 2, for surgeons, having published a clinical study in journals with an average impact factor of 3 or above (Model 1: OR = 0.655 (0.491, 0.874); Model 2: OR = 0.708 (0.531, 0.946)) and 5 or above (Model 1: OR = 0.494 (0.290, 0.843); Model 2: OR = 0.545 (0.312, 0.953)) was statistically significantly associated with lower 30-day patient readmission rates. In contrast to our findings for internists, a positive association was statistically significantly found between having a basic science publication in journals with an average impact factor of 3 or above and patients’ risk of readmission in Model 1 (OR = 1.247 (1.074, 1.447)) and Model 2 (OR = 1.230 (1.051, 1.439)).

Table 3, Table 4, Table 5 and Table 6 present a stratified analysis for patients with chronic diseases and patients without chronic diseases. Patterns were consistent between the subgroups and the whole sample. First, we continued to find that having any publication, any clinical study, or any basic science study in an international journal was not statistically significantly associated with differences in readmission rates for either internists or surgeons.

Second, consistent with our earlier results, for internists, having published a clinical study in journals with an average impact factor of 5 or above was associated with lower 30-day patient readmission rates (for chronic disease patients: Model 1: OR = 0.752 (0.593, 0.952); Model 2: OR = 0.803 (0.627, 1.029); for non-chronic disease patients: Model 1: OR = 0.551 (0.319, 0.952); Model 2: OR = 0.510 (0.316, 0.823)).

Third, similar to our earlier results, for surgeons, having published a clinical study in journals with an average impact factor of 3 or above (for chronic disease patients: Model 1: OR = 0.675 (0.482, 0.945); Model 2: OR = 0.711 (0.504, 1.003); for non-chronic disease patients: Model 1: OR = 0.515 (0.297, 0.892); Model 2: OR = 0.622 (0.362, 1.070)) and 5 or above (for chronic disease patients: Model 1: OR = 0.572 (0.351, 0.932); Model 2: OR = 0.522 (0.315, 0.868); for non-chronic disease patients: Model 1: OR = 0.364 (0.142, 0.932)) was statistically significantly associated with lower 30-day patient readmission rates. A positive association was statistically significantly found between having a basic science publication in journals with an average impact factor of 3 or above and patients’ risk of readmission (for chronic disease patients: Model 1: OR = 1.148 (0.980, 1.345); for non-chronic disease patients: Model 1: OR = 1.316 (1.304, 1.675); Model 2: OR = 1.293 (1.006, 1.662)).

The Appendix A show other sensitivity analyses, including a multivariable regression on 30-day readmission rates using non-logistic regression (see Appendix A). We present the results without any patient-level control variables (see Appendix A) and changed the outcome to 60-day readmission rates (see Appendix A), with different combinations of control variables (see Appendix A). The patterns found in the sensitivity analyses are similar to those in the main analysis (Table 1).

## 4. Discussion

Having any first-author publications was not associated with rate of readmissions. Among internists, having clinical studies published in journals with an average impact factor of 3 or above was associated with lower readmission rate, but having basic science studies published in journals with an average impact factor of 3 or above was not associated with rate of readmissions. Among surgeons, having clinical studies published in journals with an average impact factor of 3 or above was likewise associated with lower readmission rate, but having basic science studies published in journals with a similar impact factor was associated with worse clinical outcomes.

### 4.1. Comparison with the Literature

While there is an increasing strand of literature that examines the relationship between physician characteristics and patient’s outcomes, only a few other studies have quantitatively assessed physicians’ research activity and its impact on patient outcomes. One study in the United Kingdom found that research-active trusts had lower risk-adjusted mortality for acute admissions as compared to trust that are not research-active [35]. Another study found that scholarly productivity based on journal publications is positively associated with clinical performance during residency training [36]. These studies measured research activity based on number of publications, funding, number of patients recruited for study, or hospital teaching status. Our study focuses on journals’ impact factor, which measures the quality of research. It also differentiates between different types of research using individual-level data and suggests that different types of research could be linked to different outcomes.

The positive correlation between physicians’ clinical performance and their publication output in clinical studies may be explained in several ways. First, engagement in research may drive better performance in clinical work. Physicians who conduct research may be able to apply newly discovered knowledge or cutting-edge technology in clinical settings and improve patients’ outcomes. Alternatively, they may perform better in clinical settings because their familiarity with state-of-the-art research offers greater insight into how to treat patients or because the training or experience required for conducting research improves their clinical practice. Second, it is possible that physicians have characteristics such as discipline, critical thinking, organization, and teamwork, which are linked with excellence in both scholarly and clinical pursuits [36,37]. Another possibility is that strong clinical practice drives academic publications.

While physicians who had published clinical studies are associated with lower patient readmission rate, this is not the case for physicians who focus on basic science publications. The translation of scientific discoveries more broadly into clinical practice is less direct and thus more challenging. Focusing on basic science may crowd out time that physicians would otherwise spend on clinical practice without yielding proportional benefits. Because surgeons already have a high clinical workload [22], promotion criteria that emphasize research may cause surgeons to prioritize research to the detriment of their clinical practice, harming clinical outcomes.

The difference we found between clinical studies and basic science studies shows that different types of research benefit clinical outcomes differently. While studies in both clinical science and basic science require traits such as discipline, critical thinking, organization, and teamwork, basic science studies published in journals with a high impact factor are not associated with improved clinical outcomes, suggesting these traits are not driving the difference in patient outcomes. These differential results point to the possibility that clinical research itself is driving better clinical performance.

### 4.2. Policy Implications

The results of this paper support the emphasis on clinical research in hospital practice. Physicians can enrich the quality of both research in clinical studies and clinical services. For countries to nurture excellence in both research and clinical practice, a policy focus on clinical research may be appropriate. Effectively bridging the knowledge-policy gap to support the development of evidence-based policies is also important [38]. Placing reasonable weight on publications for promotion in teaching hospitals, therefore, seems reasonable, as it leads to the promotion of strong clinicians. Regarding concerns about work–life balance, stronger support systems should be given to physicians who are active in research to help them achieve excellence in both research and practice.

Despite the positive relationship between physicians’ engagement in basic science research and patients’ readmission rates, it is important not to dismiss the value of physicians’ involvement in basic research because future medical care depends on today’s scientific research [39]. Thus, instead, we should seek to understand why this relationship occurs and develop policies to better support the work of physicians in both clinical and research settings. In addition, metrics for evaluating clinicians’ performance should preferably distinguish between the two types of research so that promotions reflect diversity in research and medical practice.

### 4.3. Strengths and Limitations

This study has several strengths worth noting. We were able to quantitatively assess physicians’ research activity, breaking it down into multiple categories based on impact factor and research area, which yielded findings that may have important policy implications. Second, because female physicians in China do not change their maiden names after marriage, we were able to obtain a particularly accurate matching of their publications.

A number of limitations should also be considered in interpreting our results. First, we do not have individual-level information about the physicians other than the information about their publication in our dataset. Attending physicians in tertiary hospitals in this city have similar ages, levels of work experience, and educational backgrounds, but it is possible that they differ systematically in other characteristics. Thus, the interpretation of this study is limited to associations and not causal relationships. Second, there is a limitation associated with analyzing only first-author publications. For example, it is possible that the second author also contributed significantly to the work and thus benefited from conducting research associated with the paper but was not recognized as the first author. This may lead to a downward bias of the result because of this measurement error problem. Third, the impact factor may not be a good proxy of the quality of individual papers or individual scientists [40,41]. In addition, other research outcomes and proxies such as grants and citations were not included in our analysis. Fourth, information on clinical outcomes is limited to readmission rates, as we do not have information on mortality. We analyzed data conditional on mortality, as we restricted our analysis to individuals who were continuously enrolled in the UEBMI during the period of study.

## 5. Conclusions

For internists, having clinical studies published in journals with a high impact factor is associated with better clinical outcomes; having basic science publications is not associated with better clinical outcomes. For surgeons, having clinical studies published in journals with a high impact factor is also associated with better clinical outcomes, but having basic science studies published in journals with a high impact factor is associated with worse clinical outcomes. These differential findings on the relationship between research and clinical practice suggest the need for a careful design of policies on research and clinical practice as opposed to a blanket rejection or encouragement of research for physicians in China.

## Figures and Tables

**Table 1 ijerph-19-09760-t001:** Association Between Physicians’ First-Author Publications and Patients’ 30-Day Readmission Rates: Internist.

	Readmission Rates		Model 1		Model 2	
Publication Type	Yes	No	Adjusted Odds Ratio	*p*	Adjusted Odds Ratio	*p*
Any publication	0.23 (0.42)	0.25 (0.43)	0.966 [0.902, 1.035]	0.330	0.995 [0.934, 1.060]	0.876
Average impact factor 3+	0.22 (0.42)	0.25 (0.43)	0.917 ** [0.846, 0.994]	0.035	0.950 [0.880, 1.026]	0.194
Average impact factor 5+	0.22 (0.41)	0.24 (0.43)	0.819 *** [0.716, 0.936]	0.003	0.859 ** [0.760, 0.972]	0.016
Any clinical study	0.24 (0.42)	0.25 (0.43)	1.004 [0.923, 1.093]	0.918	1.012 [0.937, 1.093]	0.761
Average impact factor 3+	0.19 (0.39)	0.25 (0.43)	0.786 *** [0.683, 0.905]	0.001	0.849 ** [0.740, 0.975]	0.02
Average impact factor 5+	0.19 (0.39)	0.24 (0.43)	0.699 *** [0.570, 0.858]	0.001	0.746 *** [0.602, 0.926]	0.008
Any basic science study	0.22 (0.42)	0.25 (0.43)	0.947 [0.880, 1.018]	0.139	0.967 [0.906, 1.033]	0.318
Average impact factor 3+	0.25 (0.43)	0.24 (0.43)	1.013 [0.905, 1.134]	0.821	1.007 [0.915, 1.108]	0.89
Average impact factor 5+	0.23 (0.42)	0.24 (0.43)	0.934 [0.798, 1.092]	0.389	0.950 [0.819, 1.104]	0.505

Note. Standard deviations are shown in parentheses, and 95% confidence intervals are shown in brackets. Model 1 adjusts for patient characteristics, and Model 2 further adjusts for hospital fixed effects. * *p* < 0.10; ** *p* < 0.05; *** *p* < 0.01.

**Table 2 ijerph-19-09760-t002:** Association Between Physicians’ First-Author Publications and Patients’ 30-Day Readmission Rates: Surgeon.

	Readmission Rates		Model 1		Model 2	
Publication Type	Yes	No	Adjusted Odds Ratio	*p*	Adjusted Odds Ratio	*p*
Any publication	0.18 (0.38)	0.17 (0.37)	1.064 [0.950, 1.191]	0.285	1.098 * [0.991, 1.217]	0.075
Average impact factor 3+	0.18 (0.38)	0.17 (0.37)	1.083 [0.936, 1.251]	0.284	1.084 [0.941, 1.248]	0.266
Average impact factor 5+	0.15 (0.35)	0.17 (0.38)	0.868 [0.691, 1.090]	0.223	0.912 [0.727, 1.143]	0.425
Any clinical study	0.17 (0.38)	0.17 (0.38)	0.991 [0.847, 1.161]	0.915	1.023 [0.898, 1.164]	0.736
Average impact factor 3+	0.14 (0.35)	0.17 (0.38)	0.655 *** [0.491, 0.874]	0.004	0.708 ** [0.531, 0.946]	0.019
Average impact factor 5+	0.13 (0.34)	0.17 (0.38)	0.494 *** [0.290, 0.843]	0.010	0.545 ** [0.312, 0.953]	0.033
Any basic science study	0.17 (0.38)	0.17 (0.38)	1.048 [0.944, 1.162]	0.379	1.074 [0.974, 1.185]	0.154
Average impact factor 3+	0.19 (0.39)	0.17 (0.37)	1.247 *** [1.074, 1.447]	0.004	1.230 *** [1.051, 1.439]	0.010
Average impact factor 5+	0.15 (0.35)	0.17 (0.38)	0.977 [0.786, 1.215]	0.834	0.988 [0.784, 1.245]	0.917

Note. Standard deviations are shown in parentheses, and 95% confidence intervals are shown in brackets. Model 1 adjusts for patient characteristics, and Model 2 further adjusts for hospital fixed effects. * *p* < 0.10; ** *p* < 0.05; *** *p* < 0.01.

**Table 3 ijerph-19-09760-t003:** Association Between Physicians’ First-Author Publications and Chronic Disease Patients’ 30-Day Readmission Rates: Internist.

	Readmission Rates		Model 1		Model 2	
Publication Type	Yes	No	Adjusted Odds Ratio	*p*	Adjusted Odds Ratio	*p*
Any publication	0.35 (0.48)	0.36 (0.48)	0.973 [0.908, 1.044]	0.448	0.995 [0.932, 1.062]	0.886
Average impact factor 3+	0.34 (0.47)	0.36 (0.48)	0.921 * [0.846, 1.003]	0.06	0.946 [0.871, 1.027]	0.185
Average impact factor 5+	0.34 (0.48)	0.36 (0.48)	0.834 ** [0.721, 0.964]	0.014	0.867 ** [0.754, 0.997]	0.045
Any clinical study	0.36 (0.48)	0.36 (0.48)	1.023 [0.940, 1.114]	0.6	1.025 [0.946, 1.109]	0.549
Average impact factor 3+	0.31 (0.46)	0.36 (0.48)	0.816 *** [0.701, 0.950]	0.009	0.866 * [0.744, 1.009]	0.065
Average impact factor 5+	0.31 (0.46)	0.36 (0.48)	0.752 ** [0.593, 0.952]	0.018	0.803 * [0.627, 1.029]	0.083
Any basic science study	0.34 (0.47)	0.36 (0.48)	0.960 [0.892, 1.033]	0.274	0.975 [0.911, 1.043]	0.46
Average impact factor 3+	0.36 (0.48)	0.36 (0.48)	1.019 [0.904, 1.149]	0.755	1.007 [0.908, 1.116]	0.9
Average impact factor 5+	0.35 (0.48)	0.36 (0.48)	0.956 [0.805, 1.136]	0.61	0.979 [0.829, 1.156]	0.802

Note. Standard deviations are shown in parentheses, and 95% confidence intervals are shown in brackets. Model 1 adjusts for patient characteristics, and Model 2 further adjusts for hospital fixed effects. * *p* < 0.10; ** *p* < 0.05; *** *p* < 0.01.

**Table 4 ijerph-19-09760-t004:** Association Between Physicians’ First-Author Publications and Chronic Disease Patients’ 30-Day Readmission Rates: Surgeon.

	Readmission Rates		Model 1		Model 2	
Publication Type	Yes	No	Adjusted Odds Ratio	*p*	Adjusted Odds Ratio	*p*
Any publication	0.33 (0.47)	0.31 (0.46)	1.045 [0.933, 1.169]	0.449	1.090 [0.971, 1.224]	0.144
Average impact factor 3+	0.33 (0.47)	0.32 (0.47)	1.056 [0.911, 1.223]	0.469	1.053 [0.906, 1.224]	0.504
Average impact factor 5+	0.34 (0.47)	0.32 (0.47)	1.040 [0.795, 1.362]	0.774	1.045 [0.791, 1.382]	0.755
Any clinical study	0.32 (0.47)	0.32 (0.47)	0.974 [0.843, 1.125]	0.719	1.016 [0.885, 1.167]	0.82
Average impact factor 3+	0.28 (0.45)	0.32 (0.47)	0.675 ** [0.482, 0.945]	0.022	0.711 * [0.504, 1.003]	0.052
Average impact factor 5+	0.25 (0.43)	0.32 (0.47)	0.572 ** [0.351, 0.932]	0.025	0.522 ** [0.315, 0.868]	0.012
Any basic science study	0.32 (0.47)	0.32 (0.47)	1.016 [0.907, 1.138]	0.785	1.046 [0.932, 1.173]	0.446
Average impact factor 3+	0.34 (0.47)	0.32 (0.47)	1.148 * [0.980, 1.345]	0.088	1.130 [0.955, 1.336]	0.154
Average impact factor 5+	0.33 (0.47)	0.32 (0.47)	1.030 [0.783, 1.355]	0.833	1.038 [0.780, 1.383]	0.796

Note. Standard deviations are shown in parentheses, and 95% confidence intervals are shown in brackets. Model 1 adjusts for patient characteristics, and Model 2 further adjusts for hospital fixed effects. * *p* < 0.10; ** *p* < 0.05; *** *p* < 0.01.

**Table 5 ijerph-19-09760-t005:** Association Between Physicians’ First-Author Publications and Non-Chronic Disease Patients’ 30-Day Readmission Rates: Internist.

	Readmission Rates		Model 1		Model 2	
Publication Type	Yes	No	Adjusted Odds Ratio	*p*	Adjusted Odds Ratio	*p*
Any publication	0.17 (0.38)	0.21 (0.41)	0.924 [0.818, 1.044]	0.203	0.976 0.870, 1.095]	0.679
Average impact factor 3+	0.18 (0.38)	0.20 (0.40)	0.865 * [0.742, 1.008]	0.063	0.927 [0.801, 1.072]	0.307
Average impact factor 5+	0.18 (0.39)	0.20 (0.40)	0.817 [0.623, 1.072]	0.145	0.851 [0.662, 1.095]	0.210
Any clinical study	0.17 (0.38)	0.21 (0.40)	0.920 [0.793, 1.067]	0.270	0.954 [0.833, 1.091]	0.490
Average impact factor 3+	0.14 (0.34)	0.20 (0.40)	0.624 *** [0.438, 0.891]	0.009	0.704 ** [0.517, 0.959]	0.026
Average impact factor 5+	0.14 (0.35)	0.20 (0.40)	0.551 ** [0.319, 0.952]	0.033	0.510 *** [0.316, 0.823]	0.006
Any basic science study	0.16 (0.37)	0.21 (0.41)	0.888 * [0.783, 1.008]	0.066	0.924 [0.819, 1.042]	0.197
Average impact factor 3+	0.20 (0.40)	0.20 (0.40)	0.984 [0.825, 1.174]	0.861	1.006 [0.853, 1.188]	0.939
Average impact factor 5+	0.19 (0.39)	0.20 (0.40)	0.907 [0.651, 1.264]	0.565	0.899 [0.652, 1.240]	0.516

Note. Standard deviations are shown in parentheses, and 95% confidence intervals are shown in brackets. Model 1 adjusts for patient characteristics, and Model 2 further adjusts for hospital fixed effects. * *p* < 0.10; ** *p* < 0.05; *** *p* < 0.01.

**Table 6 ijerph-19-09760-t006:** Association Between Physicians’ First-Author Publications and Non-Chronic Disease Patients’ 30-Day Readmission Rates: Surgeon.

	Readmission Rates		Model 1		Model 2	
Publication Type	Yes	No	Adjusted Odds Ratio	*p*	Adjusted Odds Ratio	*p*
Any publication	0.14 (0.35)	0.14 (0.34)	1.082 [0.892, 1.313]	0.423	1.099 [0.933, 1.295]	0.260
Average impact factor 3+	0.16 (0.36)	0.14 (0.34)	1.044 [0.822, 1.326]	0.724	1.056 [0.841, 1.326]	0.638
Average impact factor 5+	0.09 (0.29)	0.14 (0.35)	0.603 *** [0.412, 0.883]	0.009	0.740 * [0.525, 1.043]	0.085
Any clinical study	0.15 (0.35)	0.14 (0.34)	1.003 [0.772, 1.304]	0.980	1.010 [0.827, 1.233]	0.923
Average impact factor 3+	0.12 (0.33)	0.14 (0.35)	0.515 ** [0.297, 0.892]	0.018	0.622 * [0.362, 1.070]	0.086
Average impact factor 5+	0.10 (0.30)	0.14 (0.35)	0.364 ** [0.142, 0.932]	0.035	0.561 [0.215, 1.462]	0.237
Any basic science study	0.14 (0.35)	0.14 (0.34)	1.080 [0.908, 1.285]	0.387	1.127 [0.957, 1.329]	0.153
Average impact factor 3+	0.16 (0.37)	0.14 (0.34)	1.316 ** [1.034, 1.675]	0.026	1.293 ** [1.006, 1.662]	0.045
Average impact factor 5+	0.10 (0.30)	0.14 (0.35)	0.911 [0.654, 1.269]	0.582	0.955 [0.670, 1.363]	0.801

Note. Standard deviations are shown in parentheses, and 95% confidence intervals are shown in brackets. Model 1 adjusts for patient characteristics, and Model 2 further adjusts for hospital fixed effects. **p* < 0.10; ** *p* < 0.05; *** *p* < 0.01.

## Data Availability

The data that support the findings of this study are available from the Urban Employee Basic Medical Health Insurance scheme, but restrictions apply to the availability of these data, which were used under license for the current study and so are not publicly available.

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
