# Peer review of "The Association of Attending Physicians’ Publications and Patients’ Readmission Rates: Evidence from Tertiary Hospitals in China Using a Retrospective Data Analysis"

_ijerph, 2022, doi:10.3390/ijerph19159760_

Round 1

Reviewer 1 Report

Dear Authors,

Thank you very much for allowing me to review this exciting paper. The topic is relevant, but the authors must address some issues before acceptance. I have summarised my review in a few main points fully expanded in the attached file. 1) This observational study should be reported according to the STROBE guideline. 2) Some of the data seem to be routinely collected; if this is the case, the manuscript structure and reporting should be integrated following the RECORD guidelines. 3) There are issues with the title which can be easily addressed; the research questions need to be specified in the manuscript to add clarity.

Author Response

Linyan Li, ScD

Assistant Professor, City University of Hong Kong

Kowloon, Hong Kong

linyanli@cityu.edu.hk

Reviewer
International Journal of Environmental Research and Public Health

Dear Reviewer,

We have revised and hereby resubmit our paper “Association between Physicians' Publications and Patients' Readmission Rates: Evidence from Tertiary Hospitals in Tertiary Hospitals in China using a retrospective data analysis” for review for potential publication in International Journal of Environmental Research and Public Health.

We are grateful for the detailed and insightful comments made by the reviewers, and effort spent by you on this paper. We have carefully addressed your comments and concerns in this revised version and outlined the changes point by point in our letter.

We appreciate your time and consideration.

Sincerely,

Linyan Li, ScD

Author Response

(The authors gave the same response as above.)
